# Stem Taper Estimation Using Artificial Neural Networks for Nothofagus Trees in Natural Forest

**Simón Sandoval** [1,2,*] and **Eduardo Acuña** [1,2]

1   Departamento de Manejo de Bosques y Medio Ambiente, Facultad de Ciencias Forestales,
    Universidad de Concepción, Concepción 4070386, Chile
2   Laboratorio de Análisis y Modelamiento de Geo-Información, Facultad de Ciencias Forestales,
    Universidad de Concepción, Concepción 4070386, Chile
*   Correspondence: simonsandoval@udec.cl

**Abstract:** The objective of the study was to estimate the diameter at different stem heights and the tree volume of the *Nothofagus obliqua* (Mirb.) Oerst., *Nothofagus alpine* (Poepp. et Endl.) Oerst. and *Nothofagus dombeyi* (Mirb.) Oerst. trees using artificial neural networks (ANNs) and comparing the results with estimates obtained from six traditional taper functions. A total of 1380 trees were used. The ANN trained to estimate the stem diameter with the best performance generated RMSE values in the training phase of 7.5%, and 7.7% in the validation phase. Regarding taper functions, Kozak's model generated better RMSE indicators, but performed not as well as that generated by the ANN. The ANN estimation of the total volume was carried out in two phases. The first used the diameter estimation to determine the volume at one-centimeter intervals along the stem (one-phase ANN), and the second used the estimation of the one-phase ANN as an additional variable in an ANN that directly estimated the tree cumulative volume (two-phase ANN). The two-phase ANN method generated the best performance for estimating the cumulative volume in relation to one-phase ANN and the Kozak taper function, generating RMSE values for *N. obliqua*, *N. alpina* and *N. dombeyi* of 9.7%, 8.9% and 8.8%, respectively.

**Keywords:** machine learning; artificial intelligence algorithms; stem profile modeling; volume estimate

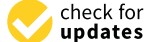



## 1. Introduction

Accuracy in volume determination at the individual tree level is critical for characterizing commercial forests. The determination of the volume requires information based on sampling methods at the tree level. However, this procedure is expensive due to the required time and investments [1]. One alternative for estimating the volume relates to indirect methods such as stem models or taper functions. The taper functions can predict the diameter at any stem height, allowing for estimating the total and the commercial volumes at any height of the tree stem [1–3]. Taper functions have become the main tool for cubing and tree bucking in forest production. They are also important aspects in sustainable forest management and planning, evaluation of forest yield, planning of forest harvesting and optimization of tree-stem bucking schemes.

Regression analysis is the most widely used parameter estimation method [4–6] to develop the parameter estimation of taper functions. However, its disadvantage is that nonlinear regression statistical inference requires strong assumptions such as homoscedasticity of variance, non-collinearity, absence of autocorrelation, among others. These strong assumptions are often arduous to meet in practical applications. Although some steps can be taken to attune the model to the assumptions, measures such as correcting heteroscedasticity and autoregressive removal also increase the complexity of building the model. If the taper function can be integrated to obtain the volume equation, it is possible to build a simultaneous system of equations between the taper function and the volume equation,

and use regression methods to obtain parameter estimates [7,8]. However, some taper functions do not satisfy the integration conditions.

Recently, artificial neural networks (ANNs) have gained importance in the area of forest biometrics, as they can provide successful predictions without any requirement for statistical assumptions and can be used effectively to solve non-linear problems [9]. In recent years, ANNs have become popular alternatives to traditional statistical methods and have been successfully applied in forest management. ANN applications are varied, and in the forest area, they have been deployed mainly to solve estimation and classification problems: in the mapping and classification of land [10], the classification of soils [11], in forest growth modeling [12,13], spatial analysis [14], prediction models for pest management and disease identification [15], species identification [16], forest land rehabilitation [17], among others. However, studies on the use of ANNs in the modeling and prediction of different characteristics of individual stands and trees are still scarce.

There are some studies where ANNs have been used to establish the models of height–diameter relationships [18,19], estimate the stem volume [1,20,21], tree heights [22], diameter distribution [23,24], and predict the stem diameter at a given height [18].

Studies that use ANNs to predict stem profile generally report the superiority of ANNs to predict stem diameter at the individual tree level in relation to the traditional taper functions. The first studies that used machine learning techniques such as individual volume and diameter estimation models were conducted 10 years ago [18,20]. In these pioneering investigations, the authors highlighted ANNs over traditional taper models, mainly because this type of model is independent of the assumptions of regression, auto-correlation of errors and has a better capacity to adapt to outliers.

Studies carried out in Brazil, such as the one by Nunes and Gorgens [25] evaluating artificial intelligence algorithms to describe the stem profile of mixed forests, concluded that one of the ANN models tested was the most accurate in relation to a Random Forest model and traditional taper functions. Additionally, in Brazil, Dolácio et al. [26] evaluated five techniques (three traditional and two non-parametric) to estimate the stem diameter and the commercial volume of *Swietenia macrophylla* Ki estimating that the ANNs are more precise than other techniques. In Turkey, Sakici and Ozdemir [27] fitted 45 kinds of ANNs and four traditional taper functions to model the stem diameter of two species, or one developed by Özçelik et al. [18], who compared four methods to predict the stem diameter of *Pinus sylvestris* L., concluded that ANNs generate the best results to estimate the stem diameter.

Recently, Socha et al. [3] generated an ANN-based stem profile model for eight species characteristic of Poland, finding the best results in a base ANN configuration compared to a traditional taper function. One result of that study indicated that there are no significant differences between the evaluated techniques, but the authors recommended the use of an ANN generated, because it presented more precise estimates in the testing dataset. In recent years, researchers have noted that there are a limited number of studies that use ANNs to make predictions of stem diameter and volume at the individual tree level, and contribute to the analysis and discussion of the potential use of new prediction techniques within the branch of artificial intelligence algorithms [3,18,28].

To date, there are no experiences of the use of ANN to describe the stem profile of native trees in Chile. Neither were these models implemented as alternatives to estimate the volume of native forests, where these ecosystems have precision difficulties in estimating the volume of individual trees, due to tree heterogeneity. On the other hand, second-growth forests of the Roble (*Nothofagus obliqua* (Mirb.) Oerst.), Raulí (*N. alpina* (Poepp et Endl.) Oerst.), and Coihue (*N. dombeyi* (Mirb.) Oerst.) forest type represents a high-value economic resource because of their fast growth and excellent timber [29,30]. Thus, it is essential to study the performance and use of these new techniques and implement them as tools that help the sustainable management and planning of natural forests in Chile. The objective of this study is to compare six traditional taper models with ANNs in different configurations

in its structure for *N. obliqua*, *N. alpina* and *N. dombeyi* trees as a model for estimating the stem diameter and the volume at different stem heights.

## 2. Materials and Methods

### 2.1. Study Area

The Roble–Raulí–Coihue forest type extends between Maule Region (35°31′ S and 71°42′ W) and Los Lagos Region (41°28′ S and 72°55′ W), between 100 and 1000 m a.s.l. in both mountain ranges, particularly in the interior slopes and mountain valleys. The second-growth forests that belong to this vegetal association of high economic interest are grouped as the forest subtype "second growth and secondary pure forests", which, according to the Native Vegetation Cadastre, cover 1468 million ha, of which 66,068 ha are managed by the National Service of Protected Wild Areas of the State [31].

The stands for sampling were selected based on representative criteria of species, thus, increasing the probability that stands will have better structure and coverage, by the means of data from CONAF [31]. All stands were relatively homogeneous in terms of species composition, anthropogenic alteration and were bigger than 10 ha. In each stand, three circular plots of 500 m$^2$ were randomly established. In each plot, the diameter at breast height of all trees ($D \geq 10$ cm) was measured, ruling out trees of other species. The three plots were processed to generate an average stand table, which was used as the basis for the selection of the sample trees in each stand. In each mixed stand, five trees of each species (*N. obliqua*, *N. alpina*, *N. dombeyi*) were selected. In monospecific stands, 10 trees of each species present were selected. All trees were destructively sampled.

The database was structured based on a sampling carried out under the framework of this study and data from other studies. The data were collected in the natural distribution range of *N. obliqua*, *N. alpina* and *N. dombeyi* second-growth forests. The base includes a total of 1380 trees with measurements of total height ($H$), diameter at breast height ($D$) and diameters with and without bark along the stem (Table 1). Total volume without bark was determined from the data. The volume without bark for each stem section was determined by the Smalian formula [32], i.e., $V_i = \left( \frac{A_1 + A_2}{2} \right) L$, where $V_i$ is the volume without bark of each section of the stem (m$^3$); $L$ is the length of the stem section (m); $A_1$ and $A_2$ are the areas at the base and up of each stem section without bark (m$^2$), respectively. The total volume of each tree was obtained from the sum of the volumes of each stem section from a height of 0.3 m to the top of the tree.

**Table 1.** Information of the trees sampled.

| Species | $n$ | Diameter (cm) | | | | Height (m) | | | | Volume (m$^3$ tree$^{-1}$) | | | |
|---|---|---|---|---|---|---|---|---|---|---|---|---|---|
| | | **Av** | **SD** | **Min** | **Max** | **Av** | **SD** | **Min** | **Max** | **Av** | **SD** | **Min** | **Max** |
| *N. obliqua* | 635 | 24.3 | 9.4 | 4.9 | 54.7 | 20.5 | 5.6 | 6.0 | 36.8 | 0.512 | 0.444 | 0.006 | 3.138 |
| *N. alpina* | 459 | 24.5 | 8.9 | 4.6 | 52.0 | 21.7 | 4.6 | 7.0 | 33.4 | 0.529 | 0.383 | 0.007 | 2.368 |
| *N. dombeyi* | 286 | 22.7 | 10.0 | 4.9 | 53.2 | 19.7 | 6.0 | 5.6 | 33.7 | 0.481 | 0.500 | 0.007 | 2.839 |

*n*: denotes the number of trees sampled, Av: the average, SD: the standard deviation, Min and Max: the minimum and maximum values, respectively.

### 2.2. Taper Models

Six models traditionally used in other studies were considered in the modeling of the stem profile. The models are detailed in Table 2 and ordered according to the year of publication.

In Table 2, $d_i$ is the diameter without bark measured in variable height $h_i$, $D$ is the diameter and $H$ is the total height of the tree. Where $Hr_i = \frac{h_i}{H}$, $X_i = \frac{H - h_i}{H - 1.3}$, $Z_i = \frac{(1 - Hr_i)^{\frac{1}{2}}}{\left(1 - \frac{1.3}{H}\right)^{\frac{1}{3}}}$, and $\beta_{1,2,\ldots 8}$ are the parameters to be estimated from the models.

**Table 2.** Taper models tested.

| Model | Model Structure | Cite |
|:---:|:---|:---:|
| 1 | $d_i = \left[ D^2 \left( \beta_1 X_i^{1.5} + \beta_2 X_i^3 + \beta_3 X_i^{32} \right) \right]^{0.5}$ | Bruce et al. [33] |
| 2 | $d_i = \left[ D^2 \left( \beta_1 X_i^{1.5} + \beta_2 X_i^{32} + \beta_3 X_i^{40} \right) \right]^{0.5}$ | Bruce et al. [33] |
| 3 | $d_i = D^2 \left[ 10^{2\beta_1} D^{2\beta_2 - 2} H^{2\beta_3} (H - h_i)^{2\beta_4} \right]^{0.5}$ | Demaerschalk [34] |
| 4 | $d_i = D \left\{ \beta_0 + \beta_1 \ln \left[ 1 - Hr_i^{\frac{1}{3}} \left( 1 - e^{-\frac{\beta_1}{\beta_2}} \right) \right] \right\}$ | Biging [35] |
| 5 | $d_i = \beta_1 D^{\beta_2} (1 - Hr)^{\beta_3 Hr_i + \beta_4 Hr_i + \beta_5}$ | Lee et al. [36] |
| 6 | $d_i = \beta_1 D^{\beta_2} + \beta_3^D Z_i^\lambda$ <br> $\lambda = \beta_4 Hr_i^2 + \beta_5 \ln(Hr_i + 0.001) + \beta_6 Hr_i^{1/2} + \beta_7 \exp(Hr_i)$ <br> $+ \beta_8 (D/H)$ | Kozak [5] |

*2.3. Modeling with Artificial Neural Networks*

The taper modeling was performed independently for the diameter without bark and for tree volume. The diameter without bark was expressed at different heights recorded in the sampling and, in the case of volume, the accumulated volume in the stump-apex direction given by each of the diameter measurements in height was considered. The cubing of each section was performed using the Smalian equation. Thus, we sought to create a trained ANN that generates estimates for the diameter and another ANN that generates estimates for the volume between two positions in the tree stem (similar to what a taper function would produce given its integration limits).

*2.4. Artificial Neural Networks Structure*

The structure of each ANN was configured separately to estimate the diameter without bark at tree height and for the commercial volume cubed in the stump-apex direction. The effect of the species was incorporated as a dummy variable in the dataset structure. The ANN configuration was performed to identify the hyper-parameters that minimized the cost function. In this study, the following hyperparameters were evaluated: number of hidden layers, number of nodes in hidden layer, activation functions and batch size. One, two and three hidden layers were evaluated, with nodes between 10, 15, 20, 25, 30 and 40 neurons. The activation functions evaluated were Rectified Linear Unit (ReLU) and Linear, and in batch sizes of 32 and 64 units. Thus, a total of 476 combinations of artificial neural network structures were performed to model the diameter without bark and the accumulated volume of the tree. The validation-split and dropout hyperparameters were maintained, and defined as 0.2 and 0.1, respectively. The supervised training of the network was carried out by randomly generating a dataset from the original one corresponding to 80% of the total data ($n$ = 1104 trees).

The back-propagation algorithm associated with the gradient-descent method was used for training in the mean square error (MSE) cost function (with an epoch of 2000 iterations). The optimizing algorithm used for all ANNs structures was Adam [37]. The ANN training used the standardized information, according to the expression $Y_i' = \frac{y_i - \mu_y}{\sigma_y}$, where $Y_i'$ is the standardized variable from its average $\mu_y$ and its standard deviation $\sigma_y$. The variables used as the input layer in the ANN that estimates the stem diameter were eight nodes in size—its variables were $D$, $H$, $h_i$, $\frac{h_i}{H}$, $\frac{H - h_i}{H - 1.3}$ and the dummy variables were relative to the three species. In the case of the ANN that estimates the accumulated volume directly, the input layer used a ninth node, which corresponded to the measurement of $d_i$. The ANN trainings were carried out with the Keras from R Interface to Keras package, version 2.3.0.0 [38].

The trained ANN was validated using the random dataset that corresponds to 20% of the data ($n$ = 276 trees). The estimates were made with the weights that minimized the MSE cost function in the $i$-th epoch; for this, the best combination of weights was saved in each

ANN structure (in a file with hdf5 extension); this allowed for the use of ANN loading only the weights obtained in the ANN structure (an example in the Appendix A).

### 2.5. Performance Metrics and Cross-Validation

To measure the predictive capabilities to estimate the diameter without bark and the volume, the performance measures used were the root mean square error (RMSE) (Equation (1)), Akaike's information criterion (AIC) (Equations (2) and (4)), and the Bayesian information criterion (BIC) (Equations (3) and (5)). The subscripts "taper" and "ANN" refer to the calculation criteria for the traditional taper functions and the ANN, respectively.

$$RMSE = \sqrt{n^{-1}SSE} \tag{1}$$

$$AIC_{taper} = n\log\left(\frac{SSE}{n}\right) + 2p \tag{2}$$

$$BIC_{taper} = n\log\left(\frac{SSE}{n}\right) + \log(n) \tag{3}$$

$$AIC_{ANN} = \log\left(\frac{SSE}{n}\right) + \frac{2m}{n-m-1} \tag{4}$$

$$BIC_{ANN} = \log\left(\frac{SSE}{n}\right) + \frac{m^2\log(n)}{n} \tag{5}$$

where $SSE$ is the sum of residual squares calculated as $\sum_{i=1}^{n}\left(Y_i - \hat{Y}_i\right)^2$, $n$ is the number of observations and $p$ is the number of parameters. According to Qi and Zhang [39], the penalizing terms $\frac{2m}{n-m-1}$ and $\frac{m^2\log(n)}{n}$ are more appropriate in ANNs and avoid the tendency to over-fit. In those terms, $m$ is the number of hidden layers between the input layer and the output layer. The validation of the fit of the taper models and the training of the ANNs was performed with cross-validation in k-fold = 10 partitions. Thus, the final result of each criterion was according to $RMSE^{(k)} = k^{-1}\sum_{j=1}^{k} RMSE_j$, $AIC^{(k)} = k^{-1}\sum_{j=1}^{k} AIC_j$, $BIC^{(k)} = k^{-1}\sum_{j=1}^{k} BIC_j$, respectively.

### 2.6. Generation of Estimated Volume

In this research, diameter estimates were generated from two methods (traditional taper functions and ANNs), and the estimated volume was generated using the same way for both methods. Stem volume is often determined using the integrated form of the cross-sectional area function estimated by the taper function of the form $V_i = k\int_{h_1}^{h_2} d_i^2 \partial h$, where $k = \pi/40,000$ produces an infinitesimal continuous estimate between two sections in stem height (between $h_1$ and $h_2$). In this study, the volume was generated using the estimated diameter in the tree stem, in consecutive sections every one cm, under the concept of the Riemann sum limit $V_i = k\int_{h_1}^{h_2} d_i^2 \partial h = k\lim_{n\to\infty}\sum_{i=1}^{n}\Delta h \cdot d_i^2$, where $\Delta h = \frac{h_2-h_1}{n}$. Thus, the volume of each section was calculated according to $V_i = \frac{1}{2}\sum_{i=1}^{n}\frac{g_i+g_{i+1}}{I_{i;i+1}}$, where $g_i$ is the basal area of the lowest section and $g_{i+1}$ is the basal area of the section immediately above and $I_{i;i+1}$ is the length of the section defined in intervals of one cm. The total estimated volume of the tree is $V = \sum_{i=1}^{n} V_i$.

### 2.7. Proof of Methods Performance

An example case was conducted to visualize the predictive ability of the taper functions and the selected ANNs. For this purpose, three trees of each species were excluded from the training and cross-validation phase, selected in each third of the diameter distribution. These trees did not participate in any fit and train process, neither for the taper functions nor in the training phase of the ANNs for the diameter without bark and the accumulated volume of each tree.

In the case of diameter, the estimates were evaluated with respect to the values obtained in each measurement and height estimates were evaluated with respect to accumulated volume in the stump-apex direction generated by Smalian formula. In this case, for the ANN estimates, a simulation of the volume was generated for each tree assigning estimates in successive sections of the stem of length intervals of one centimeter in the stump-apex direction.

## 3. Results

### 3.1. Stem Diameter Modeling

In the fit of the six traditional taper models, high precision was observed when estimating the stem diameter. According to RMSE, models 6 and 5 were the most accurate for the three species with RMSE values equal to or below 10%, with the exception of model 5 in *N. obliqua,* of which had a slightly higher RMSE value (10.4%) (Table 3). Model 6, which had the lowest RMSE, generated an estimation error of 1.54, 1.38 and 1.35 cm for *N. obliqua*, *N. alpina* and *N. dombeyi*, respectively. In all cases, the RMSE did not exceed 10.8% for the estimation of the diameter without bark, which represents error values in the estimation of the diameter of less than 1.66 cm. In Kozak's model 6, a positive bias was observed in the three species, with values between 0.002 and 0.018 cm. These bias values are also the lowest in relation to the other models. Thus, model 6 showed the best performance in estimating the stem diameter and described the shape of the trees for the three species.

The results of the modeling of the stem diameter generated by the three best ANN configurations are shown in Table 4. Even though the ANN was trained for the three species using the species class as a dummy variable, the ANN generated better results, according to the RMSE in relation to parametric models (Table 4). The ANN with the best performance generated RMSE values in the training phase of 7.5% and 7.7% for the validation phase; a lower error value, compared to 9.5%, corresponds to the best performance in the Kozak model. The error value in the estimation of the stem diameter for the testing phase in the best ANN configuration was 1.16 cm with a bias of $-0.0812$. These results were obtained in a configuration of the ANN with two hidden layers, of 30 and 25 neurons, in which the ReLU activation function was used in both layers. In this configuration, the batch side generated the best training results in relation to batch size 64, which always generated the poorest ANN performance. The number of epochs needed to ensure convergence of training was defined at 2000 cycles. In the training phase, the cost function and mean square error (MSE) stabilized approximately in 100 iterations (Figure 1). On the other hand, in the testing phase, the cost function and MSE curves decreased after 1500 cycles, and from that point, stabilization was observed, showing that the prediction capacity of the stem diameter of the ANN with new information is optimal between 1500 and 2000 training cycles.

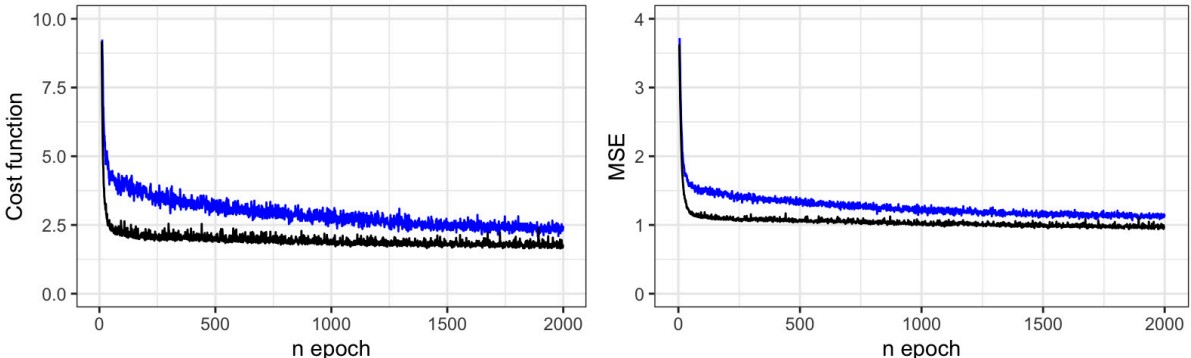

**Figure 1.** ANN training for the diameter without bark. Black denotes the training dataset and blue the validation dataset.

**Table 3.** Estimated parameters and precision fit results obtained in the traditional taper functions for the diameter without bark in the three species evaluated using the testing dataset.

| Species | Model | b1 | b2 | b3 | b4 | b5 | b6 | b7 | b8 | Bias (cm) | AIC | BIC | RMSE (cm) | RMSE (%) |
|---------|-------|----|----|----|----|----|----|----|----|-----------|-----|-----|-----------|----------|
| *N. obliqua* | 1 | 0.9793 * | −0.0079 * | 0.0013 * | | | | | | 0.082 | 6.51 | 1.61 | 1.66 | 10.8 |
| | 2 | 0.9541 * | 0.0157 * | −0.0035 * | | | | | | 0.179 | 6.49 | 1.59 | 1.64 | 10.7 |
| | 3 | 0.1240 * | 0.9806 * | −0.7845 * | 0.7204 * | | | | | 0.029 | 8.50 | 1.89 | 1.65 | 10.7 |
| | 4 | 1.1601 * | 0.3885 * | | | | | | | 0.119 | 4.48 | 1.18 | 1.62 | 10.5 |
| | 5 | 1.2676 * | 0.9473 * | 1.3716 * | −1.8155 * | 1.3268 * | | | | 0.036 | 10.48 | 2.09 | 1.61 | 10.4 |
| | 6 | 0.9850 * | 0.9917 * | 1.0001 * | 0.4059 * | −0.1256 * | 0.5402 * | −0.1274 * | 0.0208 * | 0.018 | 16.43 | 2.51 | 1.54 | 10.0 |
| *N. alpina* | 1 | 1.0015 * | −0.0195 * | 0.0015 ns | | | | | | 0.025 | 6.47 | 1.57 | 1.60 | 10.5 |
| | 2 | 0.9634 * | 0.0260 * | −0.0078 * | | | | | | 0.141 | 6.45 | 1.55 | 1.57 | 10.2 |
| | 3 | 0.1072 * | 0.9497 * | −0.7479 * | 0.7318 * | | | | | 0.023 | 8.46 | 1.85 | 1.59 | 10.4 |
| | 4 | 1.1710 * | 0.3985 * | | | | | | | 0.107 | 4.43 | 1.12 | 1.53 | 10.0 |
| | 5 | 1.3108 * | 0.9414 * | 1.9184 * | −2.4435 * | 1.5119 * | | | | 0.032 | 10.41 | 2.01 | 1.50 | 9.8 |
| | 6 | 0.8925 * | 1.0267 * | 0.9994 * | 1.0690 * | −0.2439 * | 1.3983 * | −0.7274 * | 0.1219 * | 0.002 | 16.32 | 2.40 | 1.38 | 9.0 |
| *N. dombeyi* | 1 | 0.9891 * | 0.0015 * | 0.0000 ns | | | | | | −0.044 | 6.41 | 1.51 | 1.51 | 10.5 |
| | 2 | 0.9821 * | 0.0047 * | −0.0010 * | | | | | | −0.009 | 6.41 | 1.50 | 1.50 | 10.5 |
| | 3 | 0.0957 * | 0.9604 * | −0.7890 * | 0.7760 * | | | | | 0.017 | 8.40 | 1.79 | 1.49 | 10.4 |
| | 4 | 1.1989 * | 0.4342 * | | | | | | | 0.121 | 4.42 | 1.11 | 1.52 | 10.6 |
| | 5 | 1.2683 * | 0.9504 * | 1.5661 * | −1.8257 * | 1.2954 * | | | | 0.027 | 10.36 | 1.97 | 1.43 | 10.0 |
| | 6 | 0.8236 * | 1.0669 * | 0.9982 * | 0.5046 * | −0.0820 * | 0.3046 * | −0.0852 * | 0.1123 * | 0.008 | 16.30 | 2.38 | 1.35 | 9.5 |

*: denotes the parameter significance, ns: denotes the non-significance of the parameter.

**Table 4.** Hyper-parameters obtained from the best ANN structures for the diameter without bark and the accumulated volume in the training and testing phase, 2000 epochs and activation function ReLU.

| Variable | Optimizer | Batch Size | Layer | Units | *n* pars. | Bias Test | RMSE Train | | AIC | BIC | RMSE Test | |
|---|---|---|---|---|---|---|---|---|---|---|---|---|
| | | | | | | | (cm) | (%) | | | (cm) | (%) |
| *Dsc* | Adam | 32 | 2 | 30–25 | 1071 | −0.0812 | 1.1503 | 7.5 | 16.14 | 2.22 | 1.1643 | 7.7 |
| | Adam | 32 | 2 | 30–20 | 911 | 0.0324 | 1.2508 | 8.1 | 16.22 | 2.30 | 1.2596 | 8.3 |
| | Adam | 32 | 3 | 30–25–10 | 1316 | 0.1282 | 1.2037 | 7.8 | 16.19 | 2.26 | 1.2963 | 8.5 |
| | | | | | | | ($m^3$) | (%) | | | ($m^3$) | (%) |
| *Volume* | Adam | 32 | 3 | 40–20–10 | 1441 | −0.0001 | 0.0266 | 8.2 | 14.37 | −1.43 | 0.0296 | 9.0 |
| | Adam | 32 | 3 | 40–30–20 | 2271 | 0.0008 | 0.0269 | 8.3 | 14.38 | −1.42 | 0.0318 | 9.7 |
| | Adam | 32 | 2 | 30–25 | 1101 | 0.0012 | 0.0284 | 8.8 | 14.44 | −1.36 | 0.0327 | 10.1 |

Since Kozak's model has the highest stem diameter prediction accuracy for the species studied, it was selected as the taper function used in the comparison with ANNs. To compare Kozak's model and the ANN, the fit of the models and the residuals of the diameters against the diameter observed in the three species were plotted (Figure 2). The graphs show that, for all the modeling alternatives evaluated, the errors that were normally distributed centered on zero. Kozak's model showed a tendency to higher errors for small diameters in *N. obliqua* and *N. alpina*. However, the estimation of the diameters without bark for the three species' residual distribution through Kozak's model is comparatively more dispersed in relation to the distribution of the ANN residuals, even though the neuronal network groups the three species.

*3.2. Modeling for the Accumulated Volume*

The results of the three best ANNs trained to directly estimate the volume appear in Table 4. The best performing ANN obtained RMSE of 8.2% and 9.0% in the training and testing phases, respectively. These values represent estimation errors of 0.0296 $m^3$ in the accumulated volume at the individual tree level. The bias in the testing phase was −0.0001, with this value being the one that also produced less bias in relation to the other two configurations. The resulting configuration used three hidden layers of 40, 20 and 10 neurons in each of them, using the ReLU activation function. This configuration generated 1441 weight parameters, originating from 2000 epoch training cycles. Yet, as in the ANN trained for the stem profile, the stabilization of the cost function and the MSE was achieved before 500 cycles (Figure 3). The ANN training curves show that the cost function and MSE stabilize after 500 cycles, and after that, the ANN does not show significant improvements in the testing phase. In this estimation phase, the ANN shows consistent estimates in relation to the measured volumes and the normal distribution of errors (Figure 4).

*3.3. Individual Volume Estimation*

The volume estimation capacity of the methods for each species is shown in Table 5. In general, the two-phase ANN method generated the best estimation values in relation to the one-phase ANN and the Kozak taper function. The two-phase ANN that used the estimated stem diameter in the one-phase ANN generated RMSE values for *N. obliqua*, *N. alpina* and *N. dombeyi* of 9.8%, 8.3% and 8.6%, respectively. On average, *N. alpina* species showed the lowest RMSE values, followed by *N. dombeyi* and *N. obliqua*; these results were observed in all three volume estimation methods.

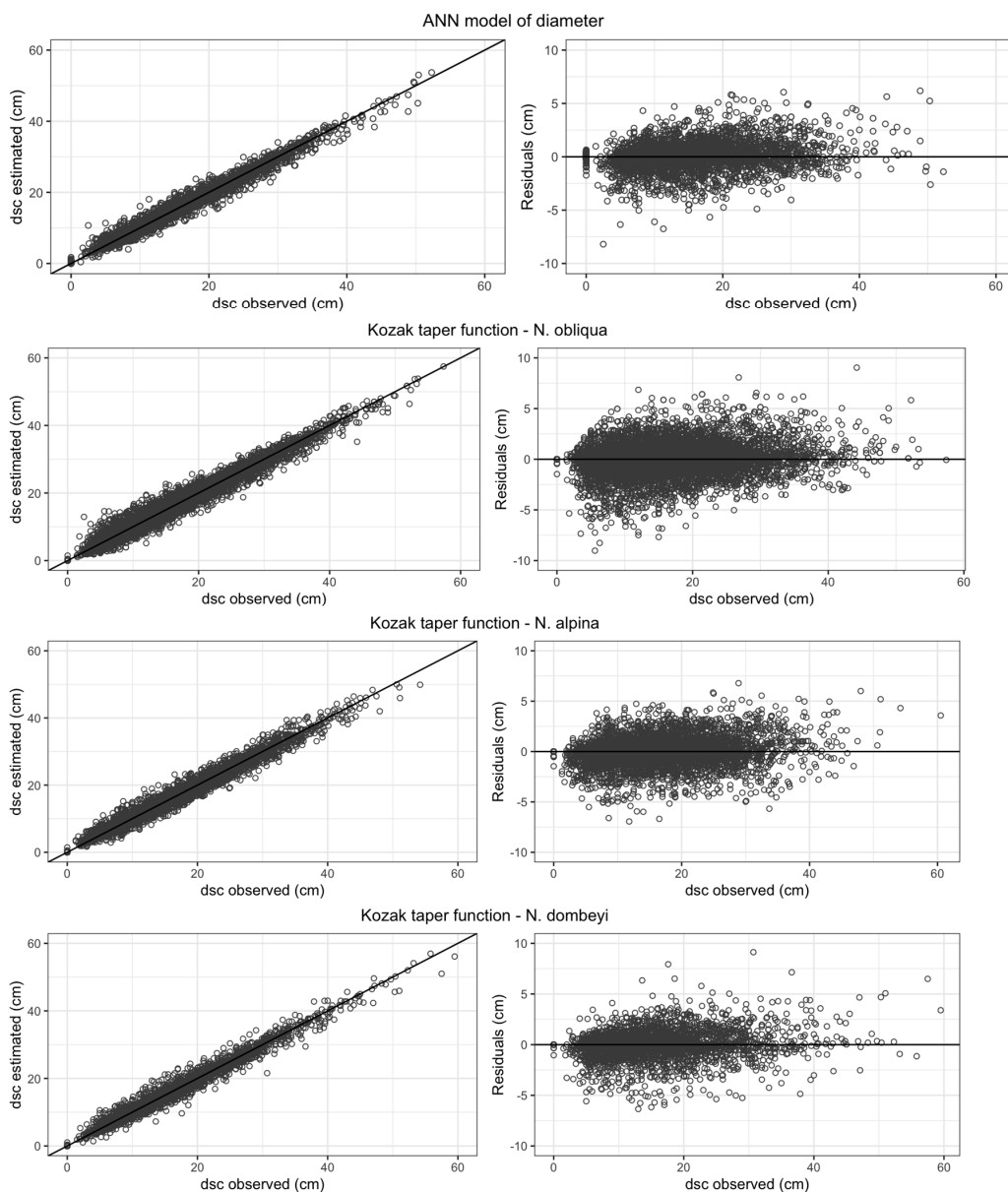

**Figure 2.** Estimation of the diameter without bark in the testing dataset using the general ANN and the taper function by Kozak [5]. For the ANN, the prediction that is made jointly for the species is shown, while the taper models were adjusted independently for each species.

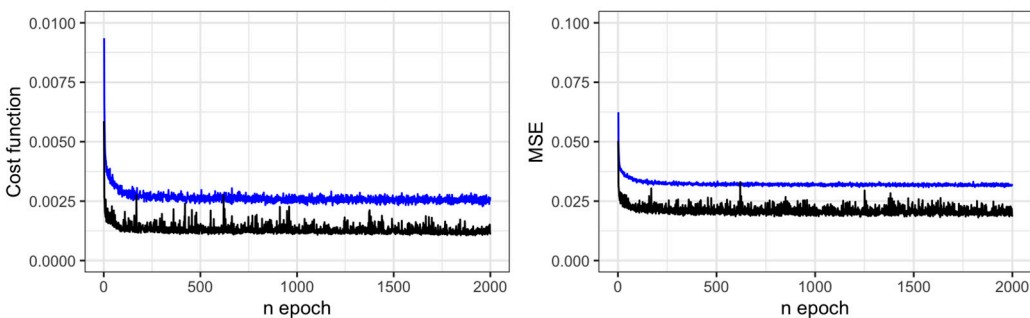

**Figure 3.** ANN training for accumulated volume without bark. Black denotes the training dataset and blue the validation dataset.

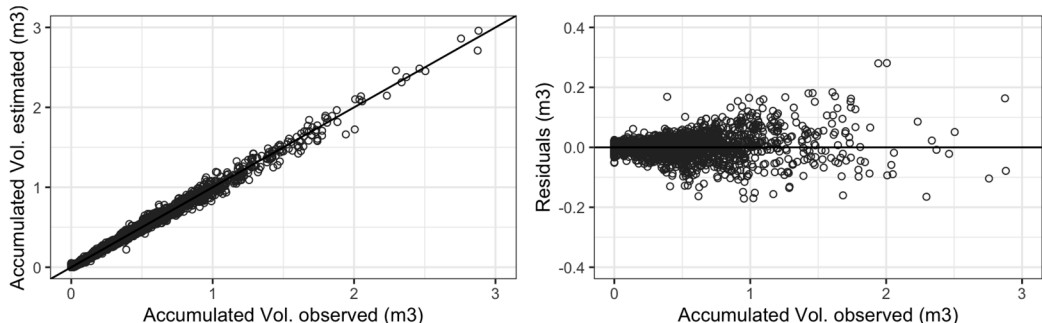

**Figure 4.** Estimation of the volume without bark accumulated in the base-up direction in the first k-fold of the dataset testing, using the ANN structure selected with the lowest RMSE.

**Table 5.** Summary of total tree volume estimation in the test basis using Kozak's taper functions, one-phase ANN and two-phase ANN of prediction. The best ranked ANNs according to the architecture shown in Table 4 were used for the calculation.

| Species | Model | RMSE | |
|---|---|---|---|
| | | $(\text{m}^3\ \text{tree}^{-1})$ | (%) |
| *N. obliqua* | Kozak | 0.0378 | 11.6 |
| | One-phase ANN | 0.0336 | 10.2 |
| | Two-phase ANN | 0.0332 | 9.8 |
| *N. alpina* | Kozak | 0.0266 | 9.9 |
| | One-phase ANN | 0.0245 | 8.9 |
| | Two-phase ANN | 0.0236 | 8.3 |
| *N. dombeyi* | Kozak | 0.0301 | 10.3 |
| | One-phase ANN | 0.0294 | 9.3 |
| | Two-phase ANN | 0.0278 | 8.6 |

### 3.4. Proof of Methods Performance

The estimate of the stem diameter for the nine trees selected from Kozak and the ANN is shown in Figure 5. In all three species, for the estimates of the stem diameter of the smaller trees corresponding to the lower third of the diameter distribution, the stem profile was accurately described with the function of Kozak and the ANNs. In general, for the three selected trees of *N. obliqua*, both prediction methods show similar estimates to the observed stem profile. In *N. alpina* in the larger diameter tree, Kozak's function underestimates the diameter, and the ANN overestimates the diameter at the basal part; however, from the middle of the height upwards, the diameter estimation is better with the ANN. In the case of *N. dombeyi*, it was observed that in the two thinnest trees, Kozak estimated the stem profile better than the ANN; however, in the thickest tree, Kozak overestimated the diameter and the best prediction was observed with the ANN.

The estimated volume of trees in the external proof is observed in Figure 6. Kozak's function did not show precise estimates in the thickest trees, evidencing problems of over-estimation of the volume in *N. obliqua* and *N. dombeyi*; in *N. alpina*, Kozak underestimated the volume of the thickest tree. The estimation of the cumulative volume with the one-phase ANN—of which the model generates the prediction of the stem diameter and then the volume is calculated—generated better estimates in relation to Kozak's model. On the other hand, the prediction made sequentially with the two-phase ANNs generated better representation of the accumulated volume compared to the two methods mentioned above. This method requires that the first ANN estimates the stem diameter (one-phase), whose value is an additional predictor in the second ANN (two-phases), and from this, the accumulated volume of the trees is estimated.

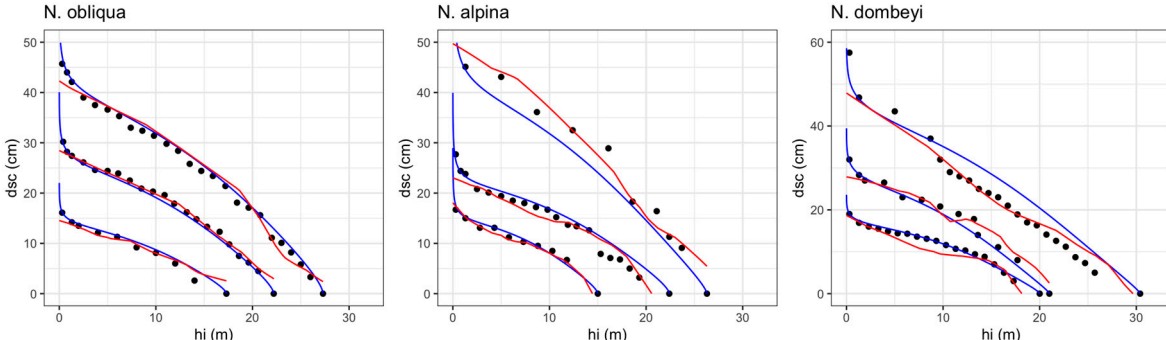

**Figure 5.** Proof for the estimation of the diameter without bark at different heights for the three selected trees of each species (dots). The blue line represents the estimate generated by Kozak's (2004) taper function and the red line represents the estimate generated by the one-phase ANN.

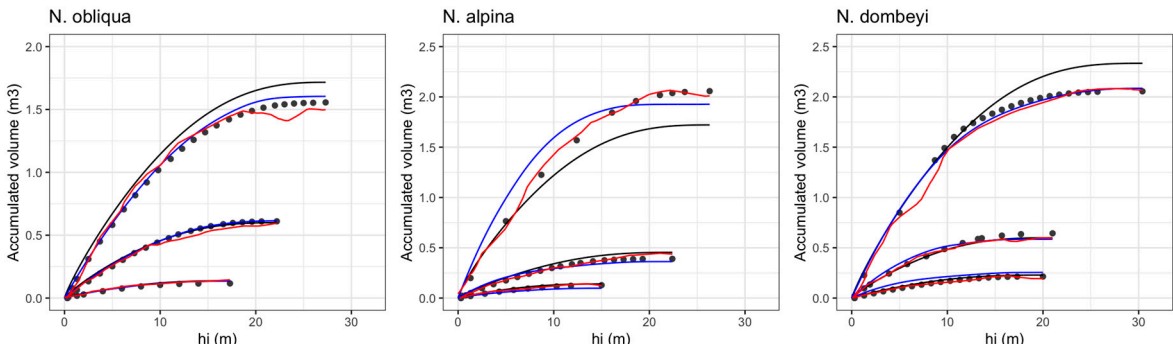

**Figure 6.** Proof for the prediction of the accumulated volume at different heights for the three selected trees of each species (dots) in the two ANN training phases. The black line represents the volume estimate generated by Kozak's (2004) taper function, the blue line represents the volume estimate generated by the one-phase ANN (diameter) and the red line represents the volume estimate generated by the two-phase ANN (diameter-volume).

## 4. Discussion

The objective of our study was to compare the performance of taper models in relation to ANNs in different architectures in trees of the species *N. obliqua*, *N. alpina* and *N. dombeyi*. In the case of the evaluation of the taper models, the Kozak [5] model generated the best results according to the RMSE, followed by the Lee et al. [36] model and by the functions of Demaerschalk [34] and Biging [35]—the latter two with similar results. Records like our results are found in Sakici and Ozdemir [27], who found that Kozak's function generated the best performance according to the RMSE in relation to four taper functions evaluated. This function has been used in numerous investigations which highlight its flexibility and ability to predict the without-bark diameter of a wide variety of forest species [25,40–42].

The base taper function of Kozak [5] was developed by Kozak [2] and has, as its main characteristic, a variable exponent that allows for accurately describing numerous shapes of the stem profile of trees. The author incorporated changes in the structure of the variable exponent and evaluated this function in 38 species, using a database of more than 53,000 trees, highlighting a low estimation bias and the option of modifying the variable exponent to improve predictive capacity [5]. However, given the complexity of its non-linear structure, a correct assignment of the value of the eight initial coefficients in the fitting process is required. Nunes and Gorgens [25] highlighted the sensitivity of the fitting to the initial assignment of the value of the coefficients. In this study, the Kozak function was evaluated for three types of forests (tropical savanna, rainforest, and semi-deciduous forest), finding the convergence of the function coefficients only in the tropical savanna forest. In our study, the assignment of the value of the initial coefficients was carried out in parts, that is, first assigning values for the parameters $\beta_1$, $\beta_2$ and $\beta_3$ in the reduced structure

$d = \beta_1 D^{\beta_2} \beta_3^D$; later, the $Z^\lambda$ variable and its associated parameters $\beta_4$, $\beta_5$, $\beta_6$, $\beta_7$ and $\beta_8$ were incorporated. In this way, our research generated the first set of parameters of Kozak's function for the use or basic assignment for new fitting in natural forests of *N. obliqua*, *N. alpina* and *N. dombeyi* in Chile.

The ANN generated better results for the prediction of the stem diameter in relation to the traditional taper functions. The three best ANNs in our study generated RMSE in the testing phases between 7.7% and 8.5%, which represents error values of diameter estimation between 1.16 and 1.29 cm (Table 3). These values were lower when compared with those generated by the Kozak function, which showed the best performance with RMSE values between 9% and 10% (1.35 and 1.54 cm) (Table 2). Previous research using ANNs to estimate the stem profile, such as that of Özçelik et al. [43], who compared the segmented taper function of Max and Burkhart [44] and ANN to estimate the stem diameter of four species in Turkey, found that ANN with a cascade correlation architecture generated the best performance with RMSE values between 4.6% and 16%. Nunes and Gorgens [25] report a similar analysis for three types of forests, indicating that the performance of an ANN was superior in relation to Random Forest models and traditional taper functions. Özçelik et al. [18] concluded that Levenberg–Marquardt's ANN model generated the best results for the estimation of the stem diameter of *Pinus sylvestris*, obtaining RMSE values between 0.91 and 1.17 cm compared to four methods of stem diameter estimation. Similar comparisons were reported in Socha et al. [3], who obtained the best performance for estimating the stem diameter of eight species in Poland using an ANN, reporting a mean RMSE value of 1.68 cm.

In general, studies that compared traditional taper functions or other non-parametric estimation algorithms in relation to ANNs, agree that the latter perform better [3,27,45,46]. However, those conclusions have been debated in another research. For example, Sanquetta et al. [47], who found that a k-Nearest Neighbors (kNN) model generated the best performance for predicting the volume of *Cryptomeria japonica* tree logs, in relation to evaluated taper models and an ANN. In a similar comparison, but evaluating individual tree volume functions, da Silva et al. [48] reported results in which the ANN performance was equivalent to that of a volume function in hybrid of *Eucalyptus grandis* and *Eucalyptus urophylla* of 6.5 years of age. Thus, most of these studies have focused on comparing the performance of traditional tapering functions versus ANNs. Our study generates a similar comparison, but also incorporates a new two-stage estimation approach using an ANN that also incorporates the species as a dummy variable.

The ANN architecture which performed best in predicting the stem diameter in our study was the two hidden layers with 30 and 25 neurons in each (Table 4). The optimizing algorithm was Adam and the activation function used in all hidden layers was ReLU. Other studies have evaluated different ANN architectures to predict stem diameter, such as Özcelik et al. [43], who found that the best results in terms of RMSE were obtained with an architecture of two or three hidden layers. Özçelik et al. [18] varied between 1 and 20 number of neurons in each hidden layer in the ANN architecture, and concluded that the optimal number of neurons in the hidden layer was between 10 and 13. Other authors concluded that increasing numbers of hidden layers improved the performance of the ANN [27], but da Silva et al. [48] recommended that the architecture of ANNs should be automatically generated by clustering algorithms.

For its part, the architecture of the ANN for the accumulated volume with the best performance in our study was of three hidden layers with 40, 20 and 10 neurons in each one. Dolácio et al. [26] generated an individual tree commercial volume model incorporating random effects, concluding that the volume can be accurately estimated using an ANN of four intermediate hidden layers, although the number of neurons in each of them is not mentioned. On the other hand, Özcelik et al. [43] argued that the number of input neurons in an ANN will depend on whether the objective is to model the stem diameter or the volume. These authors recommended simplifying the model and avoiding over-fitting problems and proposed working with 70% data training and 30% data testing. In our study,

the over-fitting problem was evaluated by partitioning the data into 80% training and 20% testing, performed cross-validation with k-fold 10, and in the training phase, 10% of nodes were randomly deactivated from a dropout layer set before the output layer. Thus, the ANN was forced to improve its generalizability in the back-propagation process of each of the epoch training cycles.

Regarding the number of training cycles or number of epochs, [27] appreciated that as the number of hidden layers increases, the performance of the ANN increases, but in this process, they also required a greater number of epochs. In our study, the training cycles required to achieve stabilization of the cost function and the MSE were 1500 for the prediction of the diameter and 500 for the accumulated volume. [18] performed 500 epochs in the training phase, finding that MSE stabilized after 200 epochs. Socha et al. [3] generated 5000 epoch cycles, but they did not refer to the minimum training cycles necessary to achieve model convergence and avoid over-fitting. Thus, knowing the number of epochs is necessary if efficient and accurate ANNs are to be generated for estimating stem diameter or accumulated volume. Therefore, Socha et al. [3] suggest that the focus of future research should be on the optimization of ANNs. In our study, 476 total combinations of ANNs were evaluated, however, the most complex ANNs—with three hidden layers and 40 neurons in each of them—did not improve the performance of the estimation in the testing phase.

To predict stem volume, ANNs were used in our study through two methods. The first used the prediction of the stem diameter for the generation of stem sections every 1 cm, in which the volume was obtained sequentially by means of the Smalian formula (one-phase ANN). The second method used the prediction of the stem diameter of the one-phase ANN as an additional input in the trained ANN to directly estimate the cumulative volume of the stem (two-phase ANN). Overall, the two-phase ANN generated cumulative volume estimates with lower RMSE relative to the Kozak [5] function and the one-phase ANN. We believe that the improvement in volume estimation was due to the one-phase ANN generating diameter predictions that served as checkpoints for the volume estimation that was generated directly from the two-phase ANN. This form of calibration has been widely used in traditional taper functions, in which a second diameter measurement is used to improve the volume estimation, often at a height greater than 1.3 m. Regarding this way of calibrating the stem profile estimates using ANNs, Özçelik et al. [18] suggest that a second diameter measurement at 60% of the total tree height is the best point to calibrate the ANN estimate. The authors mention that this measurement could be incorporated as an additional input to improve the prediction of the diameter and volume in the upper part of the tree. We trained the two-phase ANN by the means of variables frequently used in the taper functions $\left( D, \ H, \ h_i, \ \frac{h_i}{H}, \ \frac{H-h_i}{H-1.3} \right)$ plus the estimate in the one-phase ANN.

The two-phase ANN generated accurate, but more irregular, tree-level cumulative volume estimates for the same tree. This was observed in the nine external validation trees, where the cumulative volume curves estimated by the two-phase ANN presented a less smoothed shape with respect to the shape of the curve originated by the one-phase ANN. Sanquetta et al. [47] show estimates of this type in a non-smoothed way when using an ANN, generating the same effect in the volume estimation curve. These estimates, coming from the ANNs, generate predictions of the diameter or accumulated volume that could be inconsistent at the ends of the tree. Thus, at the base of the tree, the ANNs tend not to represent well the basal diameter, and at the apex of the tree, the diameter is not necessarily zero. This is a disadvantage of ANNs, in which algebraic constraints cannot be incorporated as carried out in taper functions. However, to date, most of the reported studies agree that ANNs perform better in estimating stem diameter and volume in relation to taper functions.

Recently, Socha et al. [3] concluded that ANNs are a universal tool to be used as models of stem diameter and individual tree volume. We believe that ANNs are useful tools for predicting the stem profile of trees for the species evaluated here. However, a main challenge is the implementation of ANNs in cubing or chopping systems outside the

environment or software used for training. It has been shown, both in our study and in previous ones, that ANNs generate precise diameter and volume estimates, and we believe that the main challenge should now be focused on the implementation of these models in traditional cubing and bucking systems, which so far have used the taper functions as the main cubing tool.

The biometric knowledge of the species under study can be used for the planning and design of appropriate silvicultural proposals that support a sustained use of *Nothofagus* forests, providing better estimates of products from these forests based on better information processing. This will make it possible to determine the degree of use, so that the harvest does not exceed biological growth, especially in a context of climate change, due to the fixation and sequestration of carbon in these forests. According to Donoso et al. [29], the second-growth forests of the *N. obliqua*, *N. alpina* and *N. dombeyi* forest type have been harvested for lumber for sawmills and firewood. However, one option is to increase their uses, which would allow for a significant increase in income from a forest and improvement in forest management. For the latter, the equations fitted in this paper can be used as useful management tools, allowing to know the growth and development of the stems of this forest type. Although in our paper the best taper function was Kozak's, we generated the parameterization of five other equations with a large number of tree samples. In addition, we generated a proposal of ANNs with a deep analysis of different architectures that could guide further research. One way that we believe in contributing substantially to the use and implementation of ANNs as tools for predicting the shape of *N. obliqua*, *N. alpina* and *N. dombeyi* tree stems is to make available an example of the code made in R software and the Keras library, which can be improved and used in future research.

## 5. Conclusions

In comparing the performance of the six taper functions, the Kozak [5] function consistently generated the lowest RMSE value across all three species evaluated. The one-phase ANN used to estimate the stem diameter generated better results of the RMSE in relation to the taper functions evaluated. The two-phase ANN, which directly estimates the volume, generated volume predictions with better RMSE values in relation to the estimates of the taper functions and the one-phase ANN. The best-performing one-phase ANN had a two-hidden-layer architecture with 30 and 25 neurons in the first and second layers, respectively. This architecture generated 1071 weight parameters, using the ReLU activation function and the Adam optimizer in the back-propagation process. For the case of the two-phase ANN, the best performance was observed with the three-hidden-layer architecture, with 40, 20 and 10 neurons in each layer; the ReLU activation function and Adam optimizer were also used. Regarding the number of training cycles (epochs), the convergence of the cost function and MSE in the one-phase ANN was observed after 1500 epochs, while in the two-phase ANN, it was observed after 500 epochs. The volume estimation by the means of the two-phase ANN method generated more irregular curves compared to the shape of the cumulative volume curve generated by the taper and one-phase ANN functions. Apparently, this result was generated due to the incorporation of the diameter estimated by the one-phase ANN, causing a control diameter effect guiding the estimation of the two-phase ANN, which improved the estimation of the volume with respect to the other two methods. In our study and agreeing with most of the investigations that have evaluated the capacity to estimate the stem diameter and volume using ANNs, it is concluded that these are tools with superior performance than those observed in traditional taper functions. However, we believe that the main current and future challenge should be focused on the implementation of ANNs as a cubing procedure in chopping and growth simulation systems.

**Author Contributions:** Conceptualization, S.S.; methodology, S.S.; formal analysis, S.S.; resources, S.S. and E.A.; writing—original draft preparation, S.S.; writing—review and editing S.S. and E.A. All authors have read and agreed to the published version of the manuscript.

**Funding:** This research was funded by the authors.

**Acknowledgments:** This work was supported by the Corporación Nacional Forestal with a "Fondo de Investigación del Bosque Nativo" grant N° 025/2012 "Desarrollo de herramientas de cuantificación biométrica generalizadas para el manejo y uso integral sustentable de renovales de *Nothofagus* spp.". The authors would like to acknowledge the technical support of Jorge Cancino and Carlos Valenzuela.

**Conflicts of Interest:** The authors declare no conflict of interest.

### Appendix A

Artificial neural network programming using the Keras library of the R software (training, saving the best architecture, loading weights and making predictions).

```
#===========================================================
# Setting callbacks and creating routes
name <- "model_name"
root0 <- "directory_path/Callbacks_Weights/"
dir.create(paste0(root0,name,"/"), showWarnings = FALSE)
root <- file.path(paste0(root0,name),"weights.{epoch:02d}-{val_loss:.2f}.hdf5")
# Creating Artificial Neural Network (ANN)
model <- keras_model_sequential()
# Create architecture for diameter estimates using eight predictor variables
model %>% layer_dense(units = 30, activation = "relu", input_shape = c(8)) %>%
            layer_dense(units = 25, activation = "relu", input_shape = c(8)) %>%
            layer_dropout(0.1) %>%
            layer_dense(units = 1)
# Compile ANN
modelo %>% compile(loss='mse',
                      optimizer = 'adam',
                      metrics = 'mae')
# Create Callback
cp_callback <- callback_model_checkpoint(filepath = root,
                                      save_best_only = TRUE,
                                      save_weights_only = TRUE,
                                      save_freq = "epoch",
                                      monitor = "val_loss")
# Training ANN
history <- model %>% fit(data_train,
                          y_train,
                          epochs=2000,
                          batch_size=32,
                          validation_split=0.2,
                          callbacks=list(cp_callback))
# Summary of ANN architecture
summary(model)
# Evaluation performance ANN using testing data
model %>% evaluate(data_test, y_test)
# Generating performance indices
loss <- history[["metrics"]][["loss"]]
val_loss <- history[["metrics"]][["val_loss"]]
mae <- history[["metrics"]][["mae"]]
val_mae <- history[["metrics"]][["val_mae"]]
epoch <- seq(1:length(loss))
# Setting a dataframe with performance indices
data_epochs <- as.data.frame(cbind(epoch,loss,val_loss,mae,val_mae))
# Writting a csv file with performance índices
```

```
write.csv(data_epochs, paste0("directory_path/Models_save/epochs_my_",name,".csv"),
row.names = FALSE)
# Save ANN architecture
model %>% save_model_hdf5(paste0("directory_path/Models_save/my_",name,".h5"))
# Load ANN architecture and best weights
best_file <- "weights.1860-1.95.hdf5"
root2 <- paste0("directory_path/Callbacks_Weights/",name,"/",best_file)
model <- keras_model_sequential()
model %>% layer_dense(units = 30, activation = "relu", input_shape = c(8)) %>%
                    layer_dense(units = 25, activation = "relu", input_shape = c(8)) %>%
                    layer_dense(units = 1)
model_load <- load_model_weights_hdf5(object=model, filepath = root2)
# Predictions and performance índices in new dataset testing
y_est <- model_load %>% predict(data_test_new)
error <- (y_obs - y_est)
sesgo <- mean(error)
RMSE_test <- sqrt(mean(error^2))
RMSE_test_p <- (RMSE_test/mean(y_test))*100
```

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
