# Peer review of "Stem Taper Estimation Using Artificial Neural Networks for Nothofagus Trees in Natural Forest"

_forests, doi:10.3390/f13122143_

Round 1

Reviewer 1 Report

1) Field sampling data should be described or presented as a graph, for example, the statistical distribution of tree height, diameter at breast height and volume values derived from different stands.

2) In Fig. 2, Fig.5, Fig.6, the name of “Roble”, “Rauli” and “Coihue” should be unified respectively as the same as N. obliqua, N. alpina and N. dombeyi showed in table.

3) I don’t think Fig.6 is convincing evidence for validation the two-phase ANN is better than one-phase ANN. For Roble and Coihue, the volume estimate generated by the one-phase ANN is better than two-phase ANN.

4) There are many repetitions in the introduction (such as lines 66-78) and discussion (lines 400-407), and the authors should avoid repetitive expression.

5) I don’t think the discussion in line 409-414 is meaningful, which can’t support your conclusion.

6) Many studies have proved that the ANN performed better than traditional method, in this study, I think the authors should further explore the reason, rather than cite other’s conclusion.

7) Three trees of each species were used as external validation points, the number of external validation points is too small.

8) In Table2, the data in the bias column has wrong, such as “.0.044”, “.0.009”, the authors should check carefully.

9) Line 234, “with values between 0.008 and 0.018 cm”, 0.008 should be changed as 0.002 according the table2, please verify.

10) The tenses of verbs should be consistent.

11) Line 100, 101, “The Roble-Raulí-Coihue forest type extends between Maule Region (35250 S-71400 W) and Los Lagos Region (43280 S-72560 W)”, the expression of (35250 S-71400 W) and (43280 S-72560 W) should be standardized.

Author Response

Dear Reviewer, all your suggestions and corrections were considered and incorporated in the new version of the document. Details in the attached letter. 

Kind regards, 
The authors.

Reviewer 2 Report

In this research article, the authors propose the prediction of diameter at different stem heights and the volume of three trees: Nothofagus obliqua, N. alpine, and N. dombeyi, using artificial neural networks. They evaluate the efficiency and fit of the models by modifying the number of hidden layers and the number of nodes, as well as using various taper functions.

The authors show their knowledge of the use of neural networks as well as the ins and outs of their functioning. The introduction presents a sufficient number of bibliographic references and serves as a foundation for the subsequent development of the article. The methodologies applied for obtaining the data used in the investigation and for the data processing methods utilized are also adequate. In the results, it would be convenient to better explain the parameters used to create the tables, as detailed below. The discussion shows a comparison of results with other similar research and allows showing the advances that have been reached with the use of the proposed models. Therefore, I consider that the article is suitable for publication, although I suggest that the following minor changes be made beforehand:

-Line 202. It is indicated that “Because of that the estimated volume was generated using the same way for both methods.”. However, this sentence seems incomplete…Is it a continuation of the previous sentence, or the beginning of the following one? Or is there something missing? I would suggest reviewing it.

- In table 2, there is a list of models from 1 to 6 to predict. What is the difference between those models? Are the input variables, the number of neurons or nodes, the taper functions…? Please explain it in the text of the manuscript and the table caption (if possible). Table 2 also reffers to the pareamter b1, b2…¿what are these parameters? Please, explain it in the text of the manuscript and the caption of the Table

Author Response

(The authors gave the same response as above.)

Reviewer 3 Report

It is strange that out of 1,380 trees, only 3 were used for testing. In general, it is necessary to test with about 10% of the whole data. If the RMSE and other statistics are the values ​​calculated with the validation set in the model, it is necessary to calculate them with the test data.

In addition, as this is a data-based modeling study, it is necessary to explain in detail statistical information about 1,380 trees, such as with a table. Accurate information about when and how it was acquired is necessary.

This study uses ANN for modeling to estimate the stem taper of an actual trees. Although it is a very interesting study for future information on vegetation, ANN is a model that shows well-known performance in general; so that an important part of this study is the interpretation of the data. Rather than explaining the model in detail, it is necessary to explain the data and clearly discuss the specificity of each species' results.

etc

- Scientific names should be in italics.

- Units should be written clearly. ex) m2

Author Response

(The authors gave the same response as above.)

Round 2

Reviewer 3 Report

The responses from the authors have specified that is addressed all the concerns. All the List from the comments have been well answered.